# Therapeutic Potential of Psychedelic Compounds for Substance Use Disorders

**DOI:** 10.3390/ph17111484

**Published:** 2024-11-05

**Authors:** Tamara Valdez, Valbhi Patel, Nattaphone Senesombath, Zayd Hatahet-Donovan, Mary Hornick

**Affiliations:** College of Science, Health and Pharmacy, Roosevelt University, Schaumburg, IL 60173, USA; tvaldez02@mail.roosevelt.edu (T.V.); vpatel48@mail.roosevelt.edu (V.P.); nsenesombath@mail.roosevelt.edu (N.S.); zhatahetdonovan@mail.roosevelt.edu (Z.H.-D.)

**Keywords:** psychedelics, substance use disorder, psilocybin, ketamine, lysergic acid diethylamide, 3,4-methylenedioxymethamphetamine, ayahuasca, ibogaine, peyote

## Abstract

Psychedelics have recently (re)emerged as therapeutics of high potential for multiple mental health conditions, including substance use disorders (SUDs). Despite early mid-20th century anecdotal reports and pilot studies demonstrating the possibility of these substances in efficaciously treating conditions such as alcohol and opioid use disorders, legal restrictions and social stigma have historically hindered further research into this area. Nevertheless, concurrent with the rise in SUDs and other mental health conditions, researchers have again turned their attention to these compounds, searching for differing pharmacological targets as well as more holistic treatments that might increase patient adherence and efficacy. The aim of this review is to examine the emerging evidence-based data with regards to the therapeutic treatment of SUDs with the psychedelic compounds psilocybin, ketamine, lysergic acid diethylamide (LSD), 3,4-methylenedioxymethamphetamine (MDMA), ayahuasca, ibogaine and peyote.

## 1. Introduction

Substance use disorder (SUD) is a complex physiological and psychological mental health condition that involves the usage of alcohol, cannabis, opiates, stimulants and other substances in uncontrolled amounts despite the consequences [1]. According to the 2023 National Survey on Drug Use and Health (NSDUH) conducted by the Substance Abuse and Mental Health Association (SAMHSA), 48.5 million individuals (~17% of the population) in the United States over the age of 12 were afflicted with an SUD over the previous 12 months [2]. Included within that population, 28.9 million had an alcohol use disorder (AUD), 5.7 million had an opioid use disorder (OUD), and 1.3 million had a cocaine use disorder (CUD). Additionally, 49.9 million individuals reported using nicotine products (cigarettes, cigars, vapes, etc.) within the last month (Table 1). The impact of these SUDs on the population as a whole spreads far beyond the physiological harm of the addicted individuals.

In recognizing SUDs as a public health concern, one must identify and acknowledge the risk factors and how those factors have been influenced by social, legal and governmental policies over the past 75 years. As can be seen in Table 2, individuals of lower socioeconomic status are more likely to use and/or misuse nicotine, opioids and cocaine. The only substance included which does not follow this pattern is alcohol, with the use of it increasing concurrent with total family income. In looking at AUD in particular, the highest earners still demonstrate the highest incidence, with the lowest earners next and middle-income having the lowest incidence of AUD [2]. Low socioeconomic status is associated with increased stress along with decreased access to health care, leading many to self-medicate. Hand-in-hand with the socioeconomic determinants are the data regarding racial or ethnic minorities. In nearly all cases, non-Hispanic American Indians or Alaskan Natives have the highest incidence of substance use, followed by those of two or more races (Table 2). Again, the exception to this is alcohol, in which case the non-Hispanic White population partakes at the highest percentage. The combination of low socioeconomic status with minority racial expression often fosters an environment of exposure to substances at a young age through family or peers, which in turn increases the risk of developing an SUD [3]. Additionally, having a family history of addiction can increase the risk of an individual developing an SUD by 40–60% [4]. Much of this disparity can be seen as a self-perpetuating societal issue stemming from the US War on Drugs campaign. In the 1970s, there was an increase in federal funding for drug control and treatment in part due to then-President Nixon’s claim that drug abuse was “public enemy number one” followed up in the 1980s by Nancy Reagan’s “Just Say No” campaign [5]. While these efforts to regulate drug control led to the establishment of the Drug Enforcement Agency (DEA) and allowed Congress to enact several laws like the Anti-Drug Abuse Act of 1986, the societal impact this had on the country was detrimental. Rather than bringing awareness to the issue at hand, this period in time gave leverage to those in power to label, target, criminalize and incarcerate individuals based on racial and substance abuse stereotypes [3,5]. The criminalization and incarceration associated with the War on Drugs has disproportionately impacted those of low socioeconomic status as well as Black and Indigenous People of Color (BIPOC), with the overall result being that these groups are more exposed to and subsequently likely to develop SUDs while simultaneously having less access to prevention and treatment options.

Ironically, recent advancements in the research surrounding mechanisms and potential treatments for SUDs have identified and demonstrated that one of the other targets of the War on Drugs, namely psychedelics, may be the key to a holistic therapeutic option. Whilst early experimentation in the 1950s–1960s hinted at the therapeutic potential of psychedelics such as lysergic acid diethylamide (LSD) and psilocybin (the main psychoactive component of ‘magic mushrooms’) for the treatment of SUDs, this research was practically halted when these compounds were made Schedule I substances by the 1970 Controlled Substances Act [6,7,8,9,10]. Concurrent with the rise in SUDs, particularly with the opioid epidemic and the after-effects of the COVID-19 pandemic, renewed interest has been shown in the therapeutic potential of psychedelics for the treatment of various mental health disorders [6,9,11,12,13,14,15,16]. While considering potential accessibility for marginalized populations, this review seeks to examine the emerging evidence that psychedelics may represent novel, efficacious and holistic treatment options for SUDs. In particular, we will evaluate the data surrounding psilocybin, ketamine, LSD, 3,4-methylenedioxymethamphetamine (MDMA), ayahuasca, ibogaine and peyote, focusing specifically on nicotine, alcohol, opioid and cocaine use disorders.

## 2. SUD Pathophysiology

A variety of substances may lead to addiction, from alcohol to opiates to stimulants. As measured by the 2023 NSDUH and defined by the *Diagnostic and Statistical Manual of Mental Disorders*, 5th edition, an individual may be diagnosed with an SUD based on the following criteria: the substance is recurrently taken, often in larger amounts than intended, despite its use resulting in professional, social or interpersonal problems; efforts to decrease or quit using the substance are unsuccessful, and/or more of the substance is required to achieve the desired effect; and physical and/or psychological withdrawal symptoms occur when use is stopped for a period of time [2,17]. While the specific mechanisms of action by which a substance lends itself to abuse are slightly variable, the overarching mechanism is that the substance interacts with the reward circuit, stress circuit, and/or decision-making circuits within the brain [18]. Use of a substance can stimulate the dopaminergic reward system, specifically leading to the transcription and accumulation of Delta-FosB within the neurons of the nucleus accumbens (nAC) and dorsal striatum [19,20,21]. In addition to the direct positive reinforcement from the reward circuit, the ways in which an individual responds to stress can increase the potential for drug-seeking behavior via the hypothalamus–pituitary–adrenal (HPA) axis. Although stress can create positive feelings of accomplishment or adaptability in the short term, chronic stress can lead to sustained activation of the HPA which indirectly increases dopamine release and the accumulation of Delta-FosB [22]. The inflammation caused by physiological and psychological stress is known to switch the breakdown of tryptophan off of production of serotonin and towards kynurenine [23]. Kynurenine is then metabolized to the N-methyl-D-aspartate (NMDA) agonist quinolinic acid, which leads to neurotoxicity, enhanced HPA dysregulation and glutamate excitotoxicty [24]. Indeed, glutamate imbalance has been implicated in the development of SUDs, with the metabotropic glutamate receptor 2 (mGlu2) in the prefrontal cortex (PFC), nAC and mesocorticolimbic systems exhibiting decreased function with chronic drug exposure [25]. This glutaminergic dysfunction has been shown to alter the reward pathways and increase addictive behaviors. Overall, activation of these systems can create cravings for the initiating substance, and the absence of that substance can lead to symptoms of withdrawal (Figure 1).

In reviewing individual substances, nicotine and alcohol are the most abused, likely due to their legal status when compared with opioids and cocaine. Although the above listed mechanisms underscore dependency to each of these substances, individually they do have other receptors and pathways that may be targeted when looking for therapeutics to mitigate their abuse. Nicotine, typically self-administered in the form of tobacco or vape products, stimulates the release of dopamine via its interaction with nicotinic acetylcholine receptors [26]. Alcohol use leads to a rapid increase in the release of the neurotransmitter gamma-aminobutyric acid (GABA), which, particularly with chronic use, causes downregulation of GABA type A receptors, inhibition of NMDA postsynaptic glutamate receptor activity and consequential upregulation of NMDA receptors [27]. Opioids, whether in the form of pain-relieving prescription medication or illegal heroin or synthetics, act primarily via the mu opioid receptor, with upregulation of NMDA receptors and an increase in cAMP/PKA signaling [28]. Cocaine increases dopamine in the synaptic cleft by blocking dopamine transporters, leading to increased alertness and feelings of euphoria [29]. It also inhibits serotonin and norepinephrine reuptake resulting in sympathomimetic symptoms and occasionally promoting aggressive tendencies. Withdrawal from any of these substances can cause a multitude of physical and psychological symptoms including, but not limited to, headaches, seizures, gastrointestinal distress, depression and anxiety.

## 3. Current SUD Treatments

Although there are a number of current treatments for SUDs, typically involving a combination of medication and therapy, they all generally suffer from issues with low efficacy and/or low adherence. For nicotine, those who are trying to quit can try a variety of over-the-counter nicotine replacement therapies (patches, gums, lozenges) as well as some physician-prescribed medications (e.g., bupropion and varenicline). Nevertheless, of the nearly 15 mil Americans who attempted to quit smoking in 2022, only 8.8% were successful [30]. For AUD, current treatments for acute withdrawal include benzodiazepines, gabapentin and alpha 2 agonists, with naltrexone, acamprosate and disulfiram for maintenance. As with nicotine cessation, however, 60–70% of those with AUD who attempt to quit relapse within the first 6 months to 1 year [31]. OUD is generally treated with mu opioid receptor agonists and has relapse rates ranging from 10–50% [32]. Cocaine has no specific FDA-approved medication for withdrawal or cessation although clonidine and antiemetics may be utilized during the acute phase [33]. Again, maintenance of abstinence is reportedly low, at around 25% one year post-cessation. Aside from medication, various forms of psychotherapy including cognitive behavioral therapy, individual counseling and group therapy are often recommended but do not appear to significantly increase the rates of abstinence.

## 4. Psychedelic Compounds for SUDs

A new approach is therefore needed—one which ideally encompasses a holistic approach to ensure not only mitigation of withdrawal but also the foundation and support for long-term adherence and abstinence. Building upon early, pre-scheduling research conducted on psychedelic compounds, a number of preclinical and clinical studies have again focused on the potential of these drugs to treat SUDs. Below, we highlight the emerging clinical data surrounding the use of psilocybin, ketamine, LSD, MDMA, ayahuasca, ibogaine and peyote for the treatment of nicotine, alcohol, opioid and cocaine use disorders (Table 3).

### 4.1. Psilocybin

Psilocybin (4-phosphoryloxy-N,N-dimethyltryptamine), a 5HT2A receptor agonist, is the main psychoactive component of so-called ‘magic mushrooms’. Despite the religious and health-related utilization of psilocybin-containing mushrooms by various indigenous cultures for several millennia, it was not until 1958 that Albert Hofmann isolated and determined the two main psychoactive agents—psilocybin and its active metabolite psilocin [44]. Although produced by Sandoz Pharmaceutical in 1960 under the name Indocybin and utilized in a variety of psychotolytic and psychedelic treatment models, most of the formal studies during this time focused on LSD or a combination of LSD and psilocybin or other classic psychedelics [45,46]. With the recent resurgence of interest in these compounds, however, several new studies into the potential for psilocybin to treat SUDs have been completed and more are undergoing clinical Phase I and II trials.

At present, there are thirteen active and/or currently recruiting clinical studies involving psilocybin as a treatment intervention for SUDs—nine for AUD, two for OUD and one for tobacco use disorder (TUD), with an additional six studies not yet at the recruitment phase [47]. In 2014, researchers from Johns Hopkins University School of Medicine published a pilot study of psilocybin in the treatment of tobacco addiction. Of the 15 participants, who had previously attempted to quit smoking on average six times prior, 80% were nicotine-free at the 6-month follow-up [34]. The structure of treatment involved weekly cognitive behavioral therapy (CBT) sessions with a psychologist and three spaced-out doses of 20 mg/70 kg or 30 mg/70 kg of psilocybin over a 15-week period. In a 12-month and 16+-month follow-up of the study participants, it was found that 67% and 60%, respectively, remained smoking abstinent. Interestingly, 86.7% of the participants rated the experience as one of the most personally meaningful of their lives, with 60% meeting the criteria for going through a ‘complete mystical experience’ [48,49]. Outside of the pilot study, recent surveys have indicated that those who use or have used psilocybin are at reduced odds of current nicotine dependence, and one online survey reported 38% total cessation, 28% reduced use and 34% temporary relapses to smoking following ingestion of psilocybin or another classic psychedelic [50,51]. Finally, recently posted results of the Phase II clinical study into psilocybin-facilitated smoking cessation (NCT011943994) show that, 6 months after a single dose of 30 mg/kg psilocybin combined with 13 weeks of cognitive behavioral intervention, 40.5% of those in the psilocybin group remained abstinent versus 10.0% of the group receiving CBT + standard Nicotine Replacement Therapy (NRT; transdermal nicotine patches) [47].

Comparatively, studies of psilocybin for the treatment of AUD, OUD and CUD are not as far along in the clinical pipeline, with only a few published results but significant research interest. In an early study, 31 patients with AUD were treated with alternating psilocybin and LSD sessions along with undefined psychotherapy and subsequently followed for a mean of 6 years [46]. Overall, 32% of patients remained abstinent from alcohol, while 58% had what the authors termed ‘satisfactory therapeutic effect’ (undefined). Interestingly, the LSD in this study was eventually ceased and only psilocybin sessions were used after it was determined that psilocybin had a greater effect and fewer adverse reactions. Further research into this area was halted following the Controlled Substances Act (CSA) of 1970 until the past decade, when Bogenschutz et. al. revived the inquiry. The initial pilot study composed of 10 patients with AUD involved two psilocybin sessions (21 mg/kg and 28 mg/kg) spaced 4 weeks apart along with twelve sessions of psychotherapy (preparation, motivational enhancement and debriefing) [52]. Patients only reduced alcohol intake after the first psilocybin session (week 4), and those who reported an intense trip were significantly more likely to continue to reduce or even remain abstinent from alcohol in the subsequent 28 weeks of the study. As a follow-up, the group completed a randomized, double-blind, placebo-controlled study of 95 AUD patients given either two doses of psilocybin (25 mg/70 kg and 25–40 mg/70 kg) 4 weeks apart or two doses of diphenhydramine (50 mg and 50–100 mg) along with 12 psychotherapy sessions (motivational enhancement and CBT). Results demonstrated that those receiving psilocybin had fewer heavy drinking days (9.7% vs. 23.6%) and a lower mean daily alcohol consumption than those administered diphenhydramine. While a number of other studies have called for investigating the treatment of OUD and CUD with psilocybin-assisted psychotherapy [53,54,55,56], results of clinical studies in these two SUDs remain unreported and/or in progress at this time aside from a single technical report indicating that buprenorphine and psilocybin for OUD was safely tolerated in two patients [57]. At this time, one Phase II trial on psilocybin-facilitated treatment for cocaine use has been completed (NCT02037126) and results are awaited with interest.

### 4.2. Ketamine

Ketamine, an NMDA receptor antagonist derived from phencyclidine (PCP), replaced PCP as a clinical anesthetic in the 1970’s because, while both compounds have addictive potential and can cause psychodysleptic effects, these effects were lower in ketamine while still allowing for the maintenance of hemodynamic stability during anesthesia [58,59,60]. Due to these hallucinatory side effects, ketamine has also been used in recreational contexts, although, unlike the other compounds discussed here, it is available for both clinical use and research as a Schedule III drug. As such, ketamine has been extensively researched for a variety of mental health conditions, notably depression [61,62]. As an adjunct to psychotherapy, ketamine has also been examined in a number of SUDs with or without depression as a co-morbidity. Clinical studies currently underway or recently completed include three for TUD, seven for AUD, four for OUD and one for CUD [47].

Following promising pre-clinical trials in which ketamine attenuated nicotine self-administration in rodents, several Phase I/II trials were developed to determine if this efficacy could be translated to humans with TUD [47,63]. In the only trial to have thus far released results, 10 individuals with TUD were randomly assigned to receive a single infusion of either saline or ketamine (0.5 mg/kg) over 20 min (NCT03813121). While safety seems to have been established in the reporting of one case each of non-serious adverse events (nausea, emotional distress, motor incoordination) in the week following infusion, no results with regard to abstinence or reduction in tobacco have yet been reported. The other TUD trials are in early Phase I, and we await the results with interest [47]. Evidence for the efficacy and safety of ketamine-assisted therapy in those with AUD, on the other hand, indicates potential, as reflected in the amount of current clinical trials examining this intervention (for comprehensive reviews, see [64,65]). For example, in a sample of 211 individuals receiving inpatient care for AUD, those who were administered a single intra-muscular dose of ketamine (2.5 mg/kg) alongside the standard intensive individual and group therapy reported 65.8% abstinence at one year compared with 27.0% in the control group [36]. In a recent outpatient setting, 40 individuals with AUD were treated with either a single ketamine (0.71 mg/kg) or midazolam (0.025 mg/kg) infusion paired with five weeks of motivational enhancement therapy. At the 21-day follow-up, 52.9% in the ketamine group remained abstinent versus 40.9% in the midazolam group [66]. Interestingly, the patients in this study who had a ‘mystical-type’ experience on ketamine were more likely to reduce their at-risk drinking behaviors [67]. In a recently completed trial examining the effects of ketamine or placebo with either psychotherapy or simple education on alcoholic relapse, the ketamine + psychotherapy group had the lowest amount of relapse (61.9%) while the placebo + education group had the highest (78.3%) (NCT02649231) [47]. Current trials are looking at the comparison of ketamine- versus psilocybin-assisted therapy for AUD or ketamine-assisted therapy for AUD + depression. Similar to AUD, ketamine has also shown some promise as a treatment for OUD. In fact, with regards to opioids, ketamine has the potential to not only be a treatment but also a preventative measure for OUD. A number of studies have demonstrated the benefit of utilizing low doses of ketamine as an adjunct to opioids following surgery, allowing physicians to decrease the necessary dose of opioids and thereby lower the risk of OUD development [68,69,70,71]. On the treatment side, two separate studies conducted by Krupitsky et.al. illustrated the potential of ketamine-assisted psychotherapy to treat heroin addiction. In the first study, 70 detoxified patients were divided into a low (0.2 mg/kg) and high (2.0 mg/kg) intramuscular dose of ketamine alongside psychotherapy. At 24 months post-administration, the high dose group had significantly increased levels of abstinence [38]. In the second study, single versus multiple sessions of ketamine-assisted psychotherapy for heroin addiction led to the conclusion that those receiving multiple doses were significantly more likely to remain abstinent for at least one year [72]. Current clinical trials are investigating the use of ketamine-assisted psychotherapy in OUD patients in sustained or early remission as well as those with co-morbid depression. Finally, with one Phase II trial currently active, ketamine has also been highlighted as a potential treatment for CUD. In a crossover, double-blind trial eight patients with CUD who were not actively seeking treatment or abstinence were given infusions of ketamine and lorazepam. One day after the infusions, both single and subsequent infusions of ketamine were associated with significant increases in abstinence motivation and decreases in cue-induced cravings [73]. In a subsequent trial of 20 individuals with CUD, ketamine was found to significantly reduce cocaine self-administration by 67% compared to the baseline [41]. As with other substances, the ‘mystical-type’ experiences that accompany ketamine infusion appear to be critical to its efficacy [74].

### 4.3. LSD

LSD, arguably the most iconic psychedelic, was one of the first to introduce the concept of using this class of drugs as potential treatments for SUDs. Similar to psilocybin, it is a 5HT2A receptor agonist. Synthesized by Albert Hofmann in 1938 and then subsequently forgotten about until his accidental and then iconic intentional ingestion in 1943, LSD was widely distributed to researchers and psychiatrists by Sandoz Ltd. under the name Delysid from the late 1940s–1960s. Then, as with the other classic psychedelics that had moved into the recreational sphere, it was classified as Schedule I in 1970, bringing that early research to a halt. Nevertheless, once interest began to be renewed in psychedelics as potential therapeutic agents, researchers were able to look back on those thousands of papers and anecdotes from that time period and recreate and build upon our knowledge of these compounds.

Much of the early research into LSD focused on its apparent ability to treat alcoholism anecdotally and in uncontrolled trials. The few trials that utilized control groups were inconclusive, suffering from poor design, inadequate sample sizes and questionable statistics [6,9]. A modern-day meta-analysis of these early studies, however, did find that the results were significant and consistent across seven impactful trials, with 59% of those treated with LSD demonstrating significant improvement as compared to 38% of the control group [9]. Still, likely due to the intense social stigma surrounding LSD in particular, there a no recent reports and only one ongoing clinical trial into the use of LSD for AUD (NCT05474989) [47]. Similarly, there are no direct studies utilizing LSD as an intervention for TUD, although there are some survey data. Those data, however, are complicated by self-reporting and the lack of separation between the type of psychedelic ingested and the resultant effect. While the previously mentioned online survey indicated that psychedelic use (mostly psilocybin and LSD) reduced cigarette smoking, it is unclear if one substance had more impact than any other, or indeed if participants were using more than one substance at a time [50]. Conversely, the survey utilizing data from the 2015–2019 NSDUH reported that the use of LSD was associated with increased odds of nicotine dependence [51]. With regards to OUD, one study exists from the early era of research, in which 78 male prisoners up for parole and addicted to heroin were divided into a group that received 4–6 weeks of residential treatment including one 300–500 mcg LSD session or to the standard outpatient abstinence program involving drug monitoring and weekly group therapy [39]. While the variation in therapy makes direct comparison impossible, the LSD group did report 25% abstinence at one year as compared to 5% in the standard outpatient group. While there have been calls for more studies, there are currently no active clinical trials for LSD in TUD, OUD or CUD. Indeed, a population survey using data from the 2015–2019 NSDUH indicated that LSD use was associated with increased odds of CUD [56]. Overall, despite serving as the catalyst for research into psychedelics for the treatment of SUDs, LSD has fallen by the wayside and much more research needs to be conducted to determine whether it could potentially be of benefit.

### 4.4. MDMA

MDMA is a somewhat unique poly-pharmacological psychedelic, acting not only as an agonist at 5HT2A receptors, but also as a serotonin, norepinephrine and dopamine releasing agent (SNDRA) and agonist at the trace amine-associated receptor (TAAR1). Despite not featuring in the research of the 1950s–1970s, MDMA has taken center stage in the recent psychedelic renaissance as a potential treatment for post-traumatic stress disorder (PTSD) [15,75,76]. While much of the attention is focused on MDMA-assisted therapy for PTSD, research into its potential for SUDs continues in the background.

Despite some preliminary pre-clinical data, there is no evidence indicating the usefulness of MDMA in TUD. For AUD, however, there are three current clinical trials underway testing the effects of MDMA, two of which include PTSD as a co-morbidity [47]. Details of the first open-label safety and tolerability study investigating the potential of MDMA-assisted therapy in treating those with AUD were published in a case report on the first few patients in 2019, followed by a full report in 2021 [37,77]. Whilst it was only a pilot study including 14 participants, results were favorable with patients reducing alcohol consumption by 85.7% at 9 months post-treatment and reporting increased psychosocial functioning. MDMA in this group was well tolerated with only a transient rise in blood pressure, heart rate and body temperature that returned to baseline after the session [77]. MDMA-assisted therapy is also currently being studied in a Phase II trial of co-morbid PTSD and OUD after childbirth (NCT05219175) [47]. While there have been no direct studies on the potential of MDMA-assisted therapy for CUD, this is one area where it may be contraindicated. Data from the 2015–2019 NSDUH survey indicated that those individuals who had used MDMA in their lifetime had significantly greater odds of presenting with past-year CUD (aOR 2.61; *p* < 0.001) [56]. This discrepancy could possibly be explained by the fact that both cocaine and MDMA are stimulants, leading the same population to be attracted to them, although further research is warranted. From a population perspective, one study has indicated that while Asian people were the most in favor of medical trials of MDMA-assisted therapy at 86.4%, they were the least willing to try said therapy at 40.9% [78]. Amongst other races, Hispanic/Latino people indicated the highest willingness to try MDMA-assisted therapy at 68.0%, followed by Black/AA people at 60.1%, White people at 57.5% and AIAN people at 47.5%. These percentages highlight the openness of all individuals regardless of race or ethnicity to utilize these drugs should they prove beneficial and become available.

### 4.5. Ayahuasca

Ayahuasca, an indigenous Amazonian spiritual and healing brew made from the plants *Banisteriopsis capi* and *Psychoria viridis*, is a potent mixture of the psychoactive molecule N,N-dimethyltryptamine (DMT) and a monoamine oxidase inhibitor (MAOI). Like psilocybin and LSD, DMT is considered a ‘classic psychedelic’ in that it induces psychoactive properties via agonism at the 5HT2A receptor [79]. Although somewhat less popular than mushrooms or acid, ayahuasca has made its way into Western medicine with individuals taking trips to the Amazon for what has been called ‘ayahuasca tourism’ and others incorporating it into their religious ceremonies in the U.S. itself.

Used by indigenous populations for hundreds if not thousands of years, ayahuasca has anecdotally been known to have positive effects in relieving mental health disorders, including SUDs. Only recently, however, have regulated trials or surveys been conducted to quantify this potential efficacy. A preliminary observational study conducted among a First Nations community in Canada combined four days of group counseling with two ayahuasca ceremonies, assessing psychological and behavioral patterns of SUDs before and at six months after the treatment [35]. Individuals in the study demonstrated increased hopefulness, empowerment, mindfulness and quality of life following the ayahuasca ceremonies. They also self-reported decreases from pre-ceremony in the use of tobacco (81.8% to 63.6%), alcohol (50% to 20%) and cocaine (60% to 0%), although no decline was noted in opioid use [35,55]. Regarding long-term ayahuasca use, Barbosa et. al. assessed the impact of ceremonial ayahuasca use on TUD and AUD on Brazilian members of União do Vegetal (UDV) by comparing them to a national normative sample [80]. While lifetime use of alcohol and tobacco were higher proportionally in the UDV group, the incidence of current TUD and AUD were significantly lower (*p* < 0.001 for all age ranges). Similarly, in interviews of 40 crack cocaine users in Brazil, ayahuasca was reported to create a spiritual experience, increasing their self-esteem and the socio-emotional aspects of their lives while allowing them to solve their traumas and reduce cocaine use [42]. While more intentional and regulated research needs to be done, there does appear to be some promise in the potential of ayahuasca, particularly when used in a religious/spiritual context, to reduce the severity and/or incidence of SUDs.

### 4.6. Ibogaine

Ibogaine is an indole alkaloid, isolated from the root of the *Tabernanthe iboga* plant, that has been used for centuries by the followers of the Bwiti religion in Central and Western Africa. As an integral part of their religious and initiation ceremonies, the Bwiti have reported its ability to abrogate thirst, hunger and fatigue as well as to induce a trance-like healing state [81]. As with LSD and psilocybin, ibogaine was experimented with throughout the 1960s in the U.S. and European psychedelic scene. It was during one such case of experimentation that Howard Lotsof, a soon-to-be proponent of iboga therapy, realized that it had reduced, and in some cases removed, their cravings and withdrawals from heroin [82]. Unfortunately, mainly due to its hallucinatory properties, it became illegal in 1967 and was established as a Schedule I drug following the CSA. Still, as with ayahuasca and the other drugs covered in this review, people have sought out ibogaine regardless of its legal status in hopes that it might help them with their SUDs.

Ibogaine has mostly been investigated for its potential as a treatment for OUD, and to a lesser extent CUD. While there are some pre-clinical studies indicating that ibogaine might be useful in TUD or AUD, these results have not yet been translated into human trials, although there is one current Phase II clinical trial for AUD (NCT03380728) [47,83,84,85]. Indeed, the two reports involving ibogaine use for AUD have drastically different results. In one study, a 31-year-old male with moderate AUD treated with a combination of ibogaine and 5-MeO-DMT reported that the ibogaine led to resolution of traumas behind his alcohol use while the 5-MeO-DMT induced a spiritual breakthrough. He remained abstinent from alcohol for one month [86]. In contrast, there is a second case report of a male AUD patient who died suddenly within 24 h of ibogaine treatment [87]. Indeed, there have been a small number of fatalities associated with ibogaine treatment, mostly due to its inhibition of hERG potassium channels leading to potentially fatal cardiac arrhythmias [88]. It is therefore imperative that these patients be screened and monitored properly before, during and after ibogaine treatment. Nonetheless, it has been estimated that over 10,000 patients have sought out and been treated with ibogaine worldwide in the past few decades. The vast majority of these patients seek treatment for OUD. Based on past and emerging research, it does appear that ibogaine is an extremely effective OUD treatment for reducing withdrawal and treating opioid dependence [13,40,89,90,91,92,93]. To highlight one study that followed 30 patients for 12 months post-treatment, Brown and Alper found a significant improvement in the patients’ Subjective Opioid Withdrawal Scale (SOWS) scores and reported abstinence from opioids at 1 (50%), 3 (33%), 6 (20%), 9 (37%) and 12 (23%) months [13]. Similar improvements in SOWS and an even higher rate of abstinence was reported in another study, with 75% of patients maintaining abstinence from opioids 12 months post-ibogaine therapy [40]. In patients with CUD, the intensity, frequency and duration of cravings decreased along with an improvement in depressive mood as 91.7% of patients reported that ibogaine was useful for their drug problems at three days post-treatment [43]. Given its potential benefits and considering the serious risks, it is critical that ibogaine continue to be researched and to develop optimal guidelines that govern its use in patients with an SUD.

### 4.7. Peyote

The peyote cactus (*Lophophora williamsii*) is a small, endangered cactus native to the southern portion of North America. It has been used for thousands of years by the indigenous populations of the area in religious and healing ceremonies, similar to ayahuasca and ibogaine. The main psychoactive component of the cactus is 3,4,5-trimenthoxyphenethylamine (mescaline), a phenylethylamine proto-alkaloid with properties similar to LSD and psilocybin in that it produces its hallucinogenic properties via agonism of the 5HT2A receptor [94]. Due to its endangered status as well as its current sacramental use, a vast majority of the studies performed on peyote have been retrospective or survey studies among indigenous populations. Among AI youth, more individuals reported the use of peyote for spiritual over recreational reasons, with those using it recreationally more likely to have used alcohol or cannabis within the previous thirty days [95]. In adults, those who reported use of mescaline in naturalistic settings also reported subjective improvements in depression, anxiety, AUD and general drug use disorders, with those having reported a ‘mystical experience’ indicating the greatest rates of improvement [96]. In assessing the data from the 2015–2019 NSDUH, Jones et. al. found that while peyote use did not alter the chance of nicotine dependence, it was associated with lower odds of past-year CUD (aOR 0.47; *p* < 0.05) [51,56]. As noted, cultural and environmental considerations have limited the study of peyote for medicinal value; however, there are other cacti that produce mescaline (e.g., Peruvian torch), and it may be synthesized in the lab. It would be of interest to test mescaline obtained in these ways for the potential of alleviation of SUDs.

## 5. Potential Mechanisms of Action for Psychedelics in SUD

Despite the past and emerging evidence of efficacy for the use of psychedelics as holistic treatments for SUDs, their mechanism of action in addiction is still not well understood. For the classic psychedelics (psilocybin, LSD, ayahuasca, peyote), the activation of the 5HT2A receptor is pointed to as the most likely mechanism of altering addictive behaviors. Indeed, 5HT2A and mGlu2 have been demonstrated as using crosstalk or heterodimer formation to regulate each other’s signaling pathways, thereby potentially restoring glutamate transmission within the PFC [25,97]. In restoring the PFC, glutamate transmission may result in less stress reactivity and more inhibitory control in the amygdala and dorsal raphe nuclei. This decrease in stress reactivity and subsequent stress-induced neurodegeneration at the HPA may be at least partially responsible for the reduction in drug-seeking behaviors. Ibogaine and MDMA also act via the 5HT2A receptors with some additional activity due to their unique pharmacodynamics. As an SNDRA, MDMA drastically increases the availability of not only serotonin, but also dopamine and norepinephrine, creating stimulating effects along with pleasure that, when combined with the neuroplastic effects of 5HT2A agonism, may add to its efficacy. Ibogaine, while an agonist for the 5HT2A receptors, is also an NMDA antagonist similar to ketamine. This action at the NMDA receptors may counteract the increase in the NMDA agonist quinolinic acid, helping to restore the glutamate imbalance and reset the HPA axis along with potentially leading to new synaptic formations and subsequent changes in learned behavior [81,98]. Moreover, ibogaine metabolizes in the body to noribogaine, an active metabolite that exhibits a strong affinity for mu opioid receptors. This may help to explain the reductions in opioid withdrawal symptoms and cravings associated with ibogaine treatment for OUD [93]. Beyond the pharmacologic profiles of these compounds, however, the altered consciousness and ‘mysticality’ of the experience cannot be ignored (Figure 1). In nearly all cases where it has been recorded, those individuals undergoing psychedelic therapy who have the most intense spiritual or mystical experiences on the drug are the ones who perceive the most benefit [48,99,100]. For example, in the psilocybin TUD study, it was noted that patients report feeling whole again, with the experience being likened to the opposite of an episode of PTSD [48]. Similarly, ibogaine’s oneirophrenic effects have been reported to significantly increase the overall empathy of patients along with encouraging them to learn to love themselves [101]. Ayahuasca, ibogaine and peyote, as sacraments for certain Indigenous cultures, have long been associated with spiritual enlightenment and growth [96,102,103]. Finally, the significance of the psychotherapy component cannot be ruled out at present. Although different types and intensities of therapy have been used, it is almost always a part of the experience. Determining which, indeed if any, psychotherapy is the most effective alongside psychedelics is an area requiring far more research before any potential guidelines are put into place.

## 6. Safety Considerations of Psychedelic Therapy

As these drugs make their way through various clinical trials and back into the public psyche as a whole, it is imperative that their potential harmful effects also be elucidated and acknowledged. Although the classic 5HT2A-agonist psychedelics have generally been considered reasonably non-toxic and non-addictive, there are a number of transient along with a few potential long-term adverse effects that may make their use in certain patient populations contraindicated. Specifically, psilocybin, DMT and LSD have all demonstrated a transient increase in blood pressure and heart rate during the active portion of the experience which may exclude those with certain cardiovascular diagnoses [99,104,105,106]. Additionally, these compounds are strongly contraindicated for individuals with a personal or family history of psychosis, schizophrenia and bipolar disorders as they have the potential to make these conditions manifest or worsen. Current clinical trials actively exclude such patients. Ketamine, despite its lower scheduling, has similar adverse effects as classic psychedelics as well as reports of serious gastrointestinal and urological disorders such as hemorrhagic cystitis [107]. Additionally, while more research is needed, there is some evidence that ketamine itself can be both psychologically and physically addictive [108]. MDMA, while sharing the same cardiovascular and mental health contraindications as classic psychedelics, can induce severe hyperthermia and long-term changes in liver and brain function [109,110,111]. Ibogaine’s most dangerous side effect is the activation of cardiac hERG potassium channels which can lead to QT prolongation and potentially fatal arrhythmias [88]. In addition to the above, it is likely that other potentially dangerous adverse effects as well as serious drug–drug interactions may be noted as research continues on these compounds. It is therefore extremely important that all effects be recorded and published and that patients be properly screened and evaluated prior to utilizing any of these drugs.

## 7. Conclusions

Emerging evidence has indicated the potential efficacy of psychedelic-assisted therapy for the treatment of SUDs. With increasing rates of these disorders and the number of patients who do not or cannot benefit from existing treatments, psychedelics may hold promise. Indeed, those who have undergone psychedelic-assisted treatment for an SUD, whether monitored or not, in general report improvements not only in their substance use but also in their professional and social lives. Nonetheless, far more research needs to be conducted to verify efficacy and ensure safety in developing guidelines, incorporating specific evidence-based, repeatable data. Along with this, it is imperative that those guidelines consider accessibility and equity in treatment for all SUD patients regardless of location, socioeconomic status or ethnicity.

## Figures and Tables

**Figure 1 pharmaceuticals-17-01484-f001:**
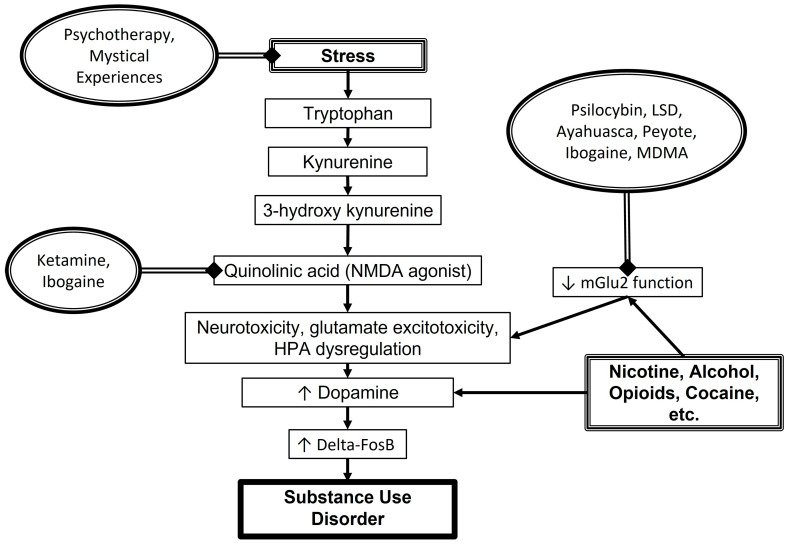
Potential mechanisms of action for psychedelic drugs in the pathophysiology of substance use disorders. LSD = lysergic acid diethylamide. MDMA = 3,4-methylenedioxymethamphetamine. NMDA = N-methyl-D-aspartate. mGlu2 = metabotropic glutamate receptor 2. HPA = hypothalamus-pituitary-adrenal axis.

**Table 1 pharmaceuticals-17-01484-t001:** Incidence of substance use among those aged 12+ in the United States in 2023. Adapted from [2].

Substance	Past Month (Mis)use	Past Year (Mis)use	Substance Use Disorder
Nicotine	49.9 mil (17.6%)	79.1 mil (27.9%)	-
Alcohol	134.7 mil (47.4%)	177.3 mil (62.5%)	28.9 mil (10.2%)
Opioids *	2.6 mil (0.9%)	9.2 mil (3.2%)	5.7 mil (2.0%)
Cocaine	1.8 mil (0.6%)	5 mil (1.8%)	1.3 mil (0.4%)

* Includes both prescription pain relievers and heroin.

**Table 2 pharmaceuticals-17-01484-t002:** Percentage of population aged 18+ using substances within the past year categorized by poverty level and race/ethnicity. NH = Non-Hispanic; AIAN = American Indian or Alaskan Native. Adapted from [2].

		Nicotine	Alcohol	Opioids	Cocaine
Poverty	Less than 100%	40.7	51.3	29.8	2.7
	100–199%	33.7	56.8	28.9	1.7
	200+%	25.4	73.8	26.5	1.8
Race or Ethnicity	NH AIAN	44.3	51.8	34.8	2.1
	NH Asian	14.3	51.9	16.2	0.8
	NH Black	31.9	62.5	30.8	1.6
	NH White	31.0	71.0	28.9	2.0
	NH Mixed	39.0	68.8	33.2	2.3
	Hispanic/Latino	25.3	62.7	23.3	2.2

**Table 3 pharmaceuticals-17-01484-t003:** Select human studies of various psychedelics for the treatment of substance use disorders (SUDs). TUD = tobacco use disorder; AUD = alcohol use disorder; OUD = opioid use disorder; CUD = cocaine use disorder.

SUD	Study	Number of Participants	Drug	Results
TUD	Johnson, et al. [34]	15	Psilocybin	80% abstinent at 6 months; 67% abstinent at 12 months
	Thomas, et al. [35]	12	Ayahuasca	Decreased tobacco use at 6 months
AUD	Bogenschutz, et al. [12]	95	Psilocybin	Fewer heavy drinking days and reduced daily alcohol consumption
	Krupitsky, et al. [36]	211	Ketamine	66% abstinent at one year
	Sessa, et al. [37]	14	MDMA	Reduced alcohol consumption by 86% at 9 months
	Thomas, et al. [35]	12	Ayahuasca	Decreased alcohol use at 6 months
OUD	Krupitsky, et al. [38]	70	Ketamine	85% abstinent at 1 month; 25% abstinent at 12 months
	Savage & McCabe [39]	78	LSD	25% abstinent at 12 months
	Brown & Alper [13]	30	Ibogaine	Reduced withdrawal symptoms; 50% abstinent at 1 month; 23% abstinent at 12 months
	Noller, et al. [40]	14	Ibogaine	Reduced withdrawal symptoms; 75% abstinent at 12 months
CUD	Dakwar, et al. [41]	20	Ketamine	Reduced self-administration by 67%
	Cruz & Nappo [42]	40	Ayahuasca	Reduced use
	Mash, et al. [43]	191	Ibogaine	Reduced intensity, frequency and duration of cravings

## Data Availability

Not applicable.

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
