# Peer review of "Therapeutic Potential of Psychedelic Compounds for Substance Use Disorders"

_pharmaceuticals, 2024, doi:10.3390/ph17111484_

Round 1
Reviewer 1 Report
Comments and Suggestions for Authors
The topic of this review is intriguing and the goal of providing scientifically correct data in such an ideological debate is commendable. The research work was detailed and the organization of the text rational. The proposed mechanisms of action are reported clearly and appropriately. The bibliography is adequate
There are a few considerations to make the work more readable:
Organize the study types and results into a couple of tables or figures to give an overview of the overall situation.
Page 10. Chapter 5 - last paragraph. After highlighting controlled and double-blind trials, their results should be tabulated separately from anecdotal or case reports. For a global evaluation of the phenomenon, in addition to the response data in abstinence from SUDs substances, it should be useful remember, as the Authors say that "the altered consciousness and 'mysticality' of the experience cannot be ignored such as the psychotherapy component" are almost always reported in the studies. These evaluations must be discussed, explored and added to the narrative of the phenomena, without forgetting that a scientific review must report the results published so far and obtained with a scientifically correct approach. As mentioned before, subjective data could also be mentioned, to better frame the typology and complexity of the recorded responses. To date, in this particular sfield we do not have sufficient evidences to formulate clinical guidelines, but only elements to suggest development lines for future research.
Therefore, both in the Conclusions and in the Abstract these reflections should be reported to make clear to the reader the need to use evidence-based data before expressing judgments of merit on the use of substances that are declared illegal in the vast majority of the readers' countries.
Author Response
Therapeutic Potential of Psychedelic Compounds for Substance Use Disorders
Response to Reviewer 1 comments:
We thank the reviewer for taking the time to review this manuscript and provide valuable feedback. Below, please find our responses to the reviewer’s comments along with instructions as to where to find the changes in the revised manuscript.
Comment 1: Organize the study types and results into a couple of tables or figures to give an overview of the overall situation.
Response 1: Published clinical studies and results have been highlighted in a new table (Table 3) in order to separate them from case studies and survey data. Figure 1 has been added to illustrate potential mechanisms of action for psychedelics in the pathophysiology of substance use disorder.
Comment 2: Page 10. Chapter 5 - last paragraph. After highlighting controlled and double-blind trials, their results should be tabulated separately from anecdotal or case reports.
Response 2: Published clinical studies and results have been highlighted in a new table (Table 3) in order to separate them from case studies and survey data.
Comment 3: For a global evaluation of the phenomenon, in addition to the response data in abstinence from SUDs substances, it should be useful remember, as the Authors say that "the altered consciousness and 'mysticality' of the experience cannot be ignored such as the psychotherapy component" are almost always reported in the studies. These evaluations must be discussed, explored and added to the narrative of the phenomena, without forgetting that a scientific review must report the results published so far and obtained with a scientifically correct approach. As mentioned before, subjective data could also be mentioned, to better frame the typology and complexity of the recorded responses.
Response 3: Specific examples have been incorporated into Section 5 of the revised manuscript to highlight the importance of this subjective data.
Comment 4: To date, in this particular field we do not have sufficient evidences to formulate clinical guidelines, but only elements to suggest development lines for future research. Therefore, both in the Conclusions and in the Abstract these reflections should be reported to make clear to the reader the need to use evidence-based data before expressing judgments of merit on the use of substances that are declared illegal in the vast majority of the readers' countries.
Response 4: Verbiage highlighting the importance of using evidence-based, repeatable data has been added to the abstract and conclusion of the revised manuscript. Additionally, a separate section (Section 6) has been incorporated into the revised manuscript describing the potential for adverse effects associated with psychedelic use.
Reviewer 2 Report
Comments and Suggestions for Authors
Research teams around the world conduct intensive research on the use of psychoactive substances in the treatment of mental disorders and now also addictions. Especially since, in the case of most addictions, despite many years of intensive research, we still do not have sufficiently effective and safe treatment methods. On this basis, I conclude that the manuscript submitted for evaluation concerns an extremely important and current problem, and the information contained therein may become a valuable for subsequent researchers planning new experiments in this field.
The manuscript is written correctly in accordance with the requirements for this type of publications. The goal is precisely defined, the substances are described in detail manner and the references are properly selected. However, in my opinion, the Authors should, describe in more detail the potential mechanism of action of the substances contained in the manuscript in the context of addiction treatment and emphasize that these substances may be associated with significant side effects. We still know little about psychedelic substances. So far, the described effects are positive, which does not change the fact that caution should be exercised, as there are still many unknowns: what doses will be safe, how often they should be administered, what the long-term effects of treatment and side effects will be. It is also necessary to assess whether it is worth using this type of treatment in the context of possible side effects.
Author Response
Therapeutic Potential of Psychedelic Compounds for Substance Use Disorders
Response to Reviewer 2 comments:
We thank the reviewer for taking the time to review this manuscript and provide valuable feedback. Below, please find our responses to the reviewer’s comments along with instructions as to where to find the changes in the revised manuscript.
Comment 1: The Authors should, describe in more detail the potential mechanism of action of the substances contained in the manuscript in the context of addiction treatment and emphasize that these substances may be associated with significant side effects.
Response 1: The potential mechanism of action for psychedelics in substance use disorders has been expanded on in the revised manuscript (Section 5) along with the addition of Figure 1 to illustrate where these substances may be acting in the pathophysiology of SUDs.
Comment 2: We still know little about psychedelic substances. So far, the described effects are positive, which does not change the fact that caution should be exercised, as there are still many unknowns: what doses will be safe, how often they should be administered, what the long-term effects of treatment and side effects will be. It is also necessary to assess whether it is worth using this type of treatment in the context of possible side effects.
Response 2: A separate section (Section 6) has been incorporated into the revised manuscript describing the potential for adverse effects associated with psychedelic use.